# Hesperidin and SARS-CoV-2: New Light on the Healthy Function of Citrus Fruits

**DOI:** 10.3390/antiox9080742

**Published:** 2020-08-13

**Authors:** Paolo Bellavite, Alberto Donzelli

**Affiliations:** 1Department of Medicine, Section of General Pathology, University of Verona Medical School, Strada Le Grazie 8, 37134 Verona, Italy; 2Medical Doctor, Scientific Committee of Fondazione Allineare Sanità e Salute, 20122 Milano, Italy; adonzelli@ats-milano.it

**Keywords:** citrus fruits, *Citrus sinensis*, hesperidin, virus and oxidative stress, COVID-19, vitamin C, SARS-CoV-2, sweet orange

## Abstract

Among the many approaches to Coronavirus disease 2019 (COVID-19) prevention, the possible role of nutrition has so far been rather underestimated. Foods are very rich in substances, with a potential beneficial effect on health, and some of these could have an antiviral action or be important in modulating the immune system and in defending cells from the oxidative stress associated with infection. This short review draws the attention on some components of citrus fruits, and especially of the orange (*Citrus sinensis*), well known for its vitamin and flavonoid content. Among the flavonoids, hesperidin has recently attracted the attention of researchers, because it binds to the key proteins of the Severe acute respiratory syndrome coronavirus 2 (SARS-CoV-2). Several computational methods, independently applied by different researchers, showed that hesperidin has a low binding energy, both with the coronavirus “spike” protein, and with the main protease that transforms the early proteins of the virus (pp1a and ppa1b) into the complex responsible for viral replication. The binding energy of hesperidin to these important components is lower than that of lopinavir, ritonavir, and indinavir, suggesting that it could perform an effective antiviral action. Furthermore, both hesperidin and ascorbic acid counteract the cell damaging effects of the oxygen free radicals triggered by virus infection and inflammation. There is discussion about the preventive efficacy of vitamin C, at the dose achievable by the diet, but recent reviews suggest that this substance can be useful in the case of strong immune system burden caused by viral disease. Computational methods and laboratory studies support the need to undertake apposite preclinical, epidemiological, and experimental studies on the potential benefits of citrus fruit components for the prevention of infectious diseases, including COVID-19.

## 1. Introduction

Coronavirus disease 2019 (COVID-19), being a new and largely unknown disease, has provided doctors with the need to investigate and try new approaches and interventions. In the early stages of the pandemic, many attempts were prompted by urgency, but now, our knowledge is increasing and consolidating. On the prevention front, various measures have been put in place, and recommendations have been issued, although not all based on rigorous evidence. Many hopes have been placed on vaccines, but their feasibility, efficacy and safety are still very uncertain. Although clinical trials are underway to test several antivirals and other agents, an important question for the population is whether there are any nutrients and food/nutrition patterns that can prevent viral infection or mitigate its severity. Diet seems to be a neglected or at least underestimated aspect, although it is acknowledged that it often plays an important role in the prevention of various diseases, even those of an infectious nature [1,2,3,4,5,6].

Among the benefits of the Mediterranean diet for the protection from many diseases, there is also the high consumption of foods rich in bioactive substances such as polyphenols and vitamins, including vitamins A, C, D, E. Food polyphenols constitute a large family of substances, with beneficial effects in a large group of communicable and non-communicable diseases. These compounds support and improve the body’s defenses against oxidative stress and in the prevention of cardiovascular diseases, atherosclerosis and cancer. In addition, they show anti-inflammatory, antiviral and antimicrobial activities. This article considers the nutraceutical properties of citrus fruits, with particular attention to hesperidin and vitamin C as potential medicines against Severe acute respiratory syndrome coronavirus 2 (SARS-CoV-2), for their activities as antiviral, antioxidants, and modulators of inflammation.

To enlighten the possible effect of the citrus components on COVID-19, it is useful to start with a brief description of the virus’s infectivity and its pathology. Figure 1 presents the main steps of the viral cycle and its consequences on the cell, with the sites where the modulating action of hesperidin and vitamin C might take place, as discussed in the subsequent chapters.

The internalization of SARS-CoV-2 is mediated by the binding of the spike glycoprotein of the virus with its receptor (ACE2) on the cell membranes. ACE2 is expressed in several tissues, including alveolar lung cells, gastrointestinal tissue, and even the brain [7,8,9,10]. The viral particle is internalized in a vesicle, whose envelope is then removed, allowing the genomic RNA to be released into the cytoplasm. The ORF1a and ORF1b RNAs are produced by the genomic RNA, and then translated into pp1a and pp1ab proteins, respectively. The pp1a and ppa1b proteins are then broken down by a proteolytic process operated by viral enzymes, resulting in a total of 16 non-structural proteins. Some non-structural proteins form a replication/transcription complex (RNA-dependent RNA polymerase), which uses genomic RNA (+) as a model. Eventually, subgenomic RNAs produced through transcription are translated into structural proteins that will form new viral particles. For this purpose, structural proteins are incorporated into the membrane and the nucleocapsid N protein combines with the positive-sense RNA, produced through the replication process, to become a nucleoprotein complex. In the Golgi endoplasmic reticulum-apparatus, the various components merge into the complete viral particle, which is finally excreted in the extracellular milieu.

The new copies of the virus spread into the environment and infect other cells and organs in the body, in a chain expansion. When the viral load is high and the cell is invaded by many viral particles, all its protein synthesis apparatus is dedicated to viral replication, up to the cell death. The last phase can take place with the “apoptosis” mechanism (if death is slow and controlled) or following an “energetic-metabolic chaos”, such as to cause the breakdown of the various cell membranes, including lysosomes, and a total loss of structural integrity. Possibly, also autoimmune phenomena are involved in the attack to the infected cell by T-lymphocytes and antibodies [11]. Eventually, both in the tissue where many cells have died (first of all in the lung), and systemically (lymph, blood, immune system, coagulation, kidney, liver), an inflammatory reaction develops, which can be clinically very serious, especially in patients with co-morbidities. Excessive and “vicious” inflammation can be mediated by a distorted activation of the cytokine network, by clotting disorders, even by a paradoxical excess of the immune reaction (autoimmunity, cytotoxic lymphocytes).

Based on this concise description, substances with a possible beneficial effect in coronavirus infection may act in various stages: (a) preventing the binding of the virus to the receptors or inhibiting the function of the receptor itself, when it sets in motion the internalization process, (b) inhibiting viral replication by blocking, for example, RNA polymerase, proteases or new particle assembly, (c) helping the cell to resist viral attack, i.e., stopping the cytotoxicity process, (d) blocking the spread of the virus in the body, (e) modulating the inflammation when, starting as an innate defensive mechanism, it becomes offensive and cytotoxic.

The prototype of the substances that act on steps (a) and (d) are specific antibodies, produced by active or passive immunization (plasma, purified IgG), even if, in the case of coronaviruses, this mechanism finds a complication in the risk of enhancement of the viral entry into target cells by the same antibodies (“antibody-dependent-enhancement”) [12] or autoimmune reactions [11]. Step (b) is the target of most antiviral drugs. Since the cytotoxicity process (step (c)) involves oxygen-derived free radicals in many cell damage mechanisms, this pathologic process could be slowed down by natural antioxidants. Finally, the modulation of the inflammatory disorders (e) can be tackled with a wide variety of steroidal and non-steroidal anti-inflammatory drugs, or with new biological agents, such as receptor antagonists or anti-cytokine antibodies.

Citrus fruits (*Rutaceae* family) are rich sources of vitamin C, anthocyanins, and flavanones, with hesperidin and naringin as the most abundant components, which have various properties, including antioxidant and anti-inflammatory activity [4,5,13]. Fibers such as pectin, more present in the solid part, help to regulate intestinal functions and hinder the absorption of LDL cholesterol. These fruits may also have beneficial effects in the prevention and treatment of viral and bacterial infections [14,15,16]. Without neglecting the best known vitamin C, here we will examine the evidence, albeit very preliminary, of a possible beneficial effect in COVID-19 of citrus fruits and their main flavonoid, hesperidin. Hesperidin is the naturally occurring form and is the glycosylated form of hesperetin, often used for its detection in plasma.

## 2. Contents and Bioavailability

*Citrus sinensis* (sweet orange) contains 0.2 g of fats, 0.7 g of proteins, 9.9 g of carbohydrates (soluble sugars), and provides 45 kcal of energy per 100 g; the fresh sweet orange juice contains traces of fats, 0.5 g of proteins, 9.8 g total carbohydrates (soluble sugars) and provides 39 kcal of energy per 100 mL [17,18]. Other active ingredients include vitamin C, carotenoids, flavanones [19,20]. Flavanones are a group of phenolic natural chemical compounds belonging to the class of flavonoids, based on the structure of the carbon atom skeleton of the flavone progenitor. Typical examples are the bitter and soluble compounds present in the peel of citrus fruits as glycosides. The most important orange flavone is hesperetin, which is found in the fruit in a glycosylated form as hesperidin. The latter is present mainly in the peel and in the white part (albedo) of citrus fruits, and consumption of the whole fruits may allow a greater intake than the juice [21]. In fact, in fresh orange juice, the content of hesperidin is about 30 mg per 100 mL, and in commercial juice it can be a little higher [22], probably because the industrial processing incorporates more peel.

There are no major differences between the different varieties of oranges and between orange, clementine and tangerine (Table 1). Moreover, lemons contain an amount of hesperidin (in mg/100 mL) comparable to that of oranges, but to drink a same volume of juice is more difficult. The flavonoid content of the red orange [23] is mainly hesperidin (43.6 mg/100 mL), followed at a distance by narirutin (4.8 mg/100 mL) and dimidine (2.4 mg/100 mL).

According to a recent review [24], the content of hesperidin in 100 mL of juice is: orange 20–60 mg, tangerines 8–46 mg, lemon 4–41 mg, grapefruit 2–17 mg. This means that we can take about 100 mg of hesperidin, just in a large glass of orange juice. Based on these data, it can be said that the choice of the most suitable fruits for a better intake of hesperidin could be made between oranges, mandarins and clementines, according to individual preferences, and costs.

Note that the higher levels of hesperidin (and of other flavonoids) are in citrus peel: flavedo and albedo. The concentration in the albedo is at least one order of magnitude greater than that of the juice [24]. Therefore, the greater intake could be obtained by preferring citrus fruits from organic farming, of which you can also eat the peel.

Some tests have assessed the amount of hesperidin (or its metabolite hesperetin) in the blood of people drinking orange juice. Healthy volunteers drank orange juice in one intake (8 mL/kg) and blood and urine samples were collected between 0 and 24 h after administration [25]. The peak plasma concentration of hesperetin was 2.2 ± 1.6 micromol/L, with significant variations in different subjects. Elimination half-life ranged from 1.3 to 2.2 h, indicating short-term kinetics. In another experiment [26], after a night fast, five healthy volunteers drank 0.5 or 1 L of commercial orange juice, containing 444 mg/L of hesperidin, along with a polyphenol-free breakfast. The flavanone metabolites appeared in the plasma 3 h after the ingestion of the juice, peaked between 5 and 7 h, then returned to the baseline value at 24 h. The peak plasma concentration of hesperetin was 0.46 ± 0.07 micromol/L and 1.28 ± 0.13 micromol/L, after taking 0.5 and 1 L, respectively. The authors concluded that, in the case of moderate or high consumption of orange juice, flavanones represent an important part of the pool of total polyphenols in plasma.

However, it would not be correct to evaluate the bioavailability of phenols only with the dosage of hesperetin. In fact, there is evidence that hesperidin and naringin are metabolized by intestinal bacteria, mainly in the proximal colon, with the formation of their aglycones, hesperetin and naringenin and various other small phenols [27]. Studies have also shown that citrus flavanones and their metabolites are able to influence the composition and activity of the microbiota, and to exert beneficial effects on gastrointestinal function and health. Other bioavailability studies have calculated that, if the phenolic catabolites derived from the colon are added to the glucuronide and sulphate metabolites, the polyphenols derived from orange juice are much more abundant and available than previously thought [28,29].

Human studies have long shown the safety and good tolerability of hesperidin up to very high doses [24]. In animal studies, hesperidin showed a good safety profile [30], with a median lethal dose (LD50) of 4837.5 mg/kg, and in chronic administration up top, 500 mg/kg of the flavanone did not induce any abnormalities in body weight, clinical signs and symptoms and blood parameters.

## 3. Hesperidin and the Virus

The discovery that the molecule of hesperidin has a chemical-physical structure suitable for binding to key proteins in the functioning of the SARS-CoV-2 virus has recently aroused scientific interest. At least six searches yielded concordant results [31,32,33,34,35,36]. The researchers started from the detailed knowledge of the virus protein structure, to ascertain which molecules, natural or artificial, are capable of binding with a low binding energy (the lower the energy required, the stronger and more specific the binding is). This technique, called “in silico”, is currently applied to predict drug behavior and accelerate the detection rate, since it allows screening many drugs, reducing the need for expensive laboratory work and limiting clinical trials to the best candidates.

Wu and collaborators [31] have tested 1066 natural substances with potential antiviral effect, plus 78 antiviral drugs already known in the literature, for their binding to SARS-CoV-2 proteins. Of all, hesperidin was the most suitable to bind to the “spike”. By superimposing the ACE2—receptor binding domain (RBD) complex on the hesperidin—RBD complex, a clear overlap of hesperidin with the ACE2 interface is observed, which suggests that hesperidin may disrupt the interaction of ACE2 with RBD.

A second theoretical site of low energy binding of hesperidin with SARS-CoV-2 is the main protease that allows the processing of the first proteins transferred from the viral genome-pp1a and pp1ab-into functional proteins in the host cell [31]. This enzyme is called “3Clpro” or “Mpro” by the various authors, and is the target of many chemical antiviral drugs. This specific binding has also been confirmed by other authors: in a screening of 1500 potential molecules capable of binding to 3CLpro, hesperidin is the second most efficient for binding to chain A, with a free energy of −10.1 kcal mol^−1^ [32]. Lopinavir (−8.0) and ritonavir (−7.9) are given as reference drugs, and they show less binding capacity. The binding to chain B occurs with −8.3 kcal mol^−1^, while lopinavir (−6.8) and ritonavir (−6.9) have lower binding capacity.

Another detailed molecular docking study of the interaction between hesperidin and Mpro was recently published [35]. In a screening of 33 natural and already known antiviral molecules, the authors found that the lower binding energy (indicating maximum affinity) is characteristic of rutin (−9.55 kcal/mol), followed by ritonavir (−9.52 kcal/mol), emetine (−9.07 kcal/mol), hesperidin (−9.02 kcal/mol), and indinavir (8.84 kcal/mol). Hesperidin binds with hydrogen bonds to various amino acids, mainly THR24, THR25, THR45, HIS4, SER46, CYS145. Further evidence came from the work by Joshi et al. [36], who identified hesperidin among several natural molecules that strongly bind to SARS-CoV-2 main protease, and interestingly also to the viral receptor angiotensin-converting enzyme 2 (ACE-2).

A research published by Indonesian authors and so far available in preprints has examined with computational methods a wide range of active principles of the medicinal plants *Curcuma* sp., *Citrus* sp. (orange), *Caesalpinia sappan* and *Alpinia galanga*, for their ability of “molecular docking” towards viral proteins [34]. For the three major proteins involved in virus infection, hesperidin was the most efficient binding molecule, with docking points of −13.51, −9.61 and −9.50 respectively to the SARS-CoV-2 protease, to the glycoprotein-RBD Spike and to the ACE2 receptor. Hesperidin performs a better interaction with the SARS-CoV-2 protease than lopinavir, a reference drug used today in the clinical trials for Covid-19. These authors have observed that, in addition to hesperidin, other orange flavonoids less represented quantitatively, like tangeretin, naringenin and nobiletine, also have a low binding energy (comparable to the reference ligands, lopinavir and nafamostat) to the three essential proteins, suggesting that these interactions could also contribute to the inhibitory effect against virus infection. According to another “molecular docking” research study [33], out of 26 natural phenolic compounds that are candidates for antiviral action, hesperidin was the one with the highest binding capacity to the crystallized form of the main protease of SARS-CoV-2. The flavanone interacts with several amino acids of the protein through hydrogen bonds, and the interaction of hesperidin is more effective than that realized by the reference drug nelfinavir (with scores of −178.59 and −147.38 respectively).

There is an important precedent when the authors studied natural compounds capable of inhibiting 3CLpro of the SARS virus [37], using cell-based proteolytic cleavage assays. Out of seven phenolic compounds tested, hemodyne and hesperetin inhibited proteolytic activity in a dose-dependent manner, with IC50 of 366 micromol/L and 8.3 micromol/L respectively. Interestingly, this research suggests that the inhibition of viral protease occurs at concentrations of hesperidin of the same order of magnitude as those achievable in plasma, with a large oral supplement of orange juice. Since coronavirus main protease structural backbone and active site conformation are conserved despite sequence variations [36], it is conceivable that the inhibitory effect of hesperidin previously observed in SARS virus can be exploited also in SARS-CoV-2.

## 4. Antioxidant Activity

An efficient oxidative metabolism at the mitochondrial level (without unwarranted formation of free radicals) and the balance of oxidation reactions, due to the intervention of the enzymatic systems and various scavenger molecules, are essential for the vitality of the cells of each tissue. Several viruses break this balance and induce oxidative stress, which in turn facilitates specific phases of the life cycle of SARS-CoV-2 [38], and eventually cell death (Figure 2). Hesperidin contributes significantly to antioxidant defense systems as an effective agent against superoxide and hydroxyl radicals [39], and its derivative hesperetin inhibits nitric oxide production by LPS-stimulated microglial cells [40].

Systemic perturbations associated with the severity of COVID-19 disease include free heme release and hyperferritinemia, a sign of dysregulation of iron metabolism, which in turn induces the production of reactive oxygen species, such as superoxide anion (O_2_^−^), hydrogen peroxide (H_2_O_2_); hydroxyl radical (·OH), and promotes the oxidative stress [41,42]. In this pathological process, the dysfunction of mitochondrial oxidative metabolism also plays an important role, leading to platelet damage and promoting the formation of thrombi [43]. Another experimental model of generation of free radicals is represented by ischemia and reperfusion (I/R). This occurrence has been described also in COVID-19 [44], or can aggravate the treatment with drugs such as chloroquine [45]: during prolonged hypoxia of a tissue, the cells undergo structural damage, especially in the mitochondria and the endoplasmic reticulum. In addition, the purine metabolism ends with the formation of abundant quantities of xanthine. When the oxygen brought by the blood returns to the same tissue, these biochemical mechanisms—i.e., the mitochondrial chain, the endoplasmic reticulum and xanthine oxidase—generate a monovalent reduction of oxygen with the formation of superoxide and other chain radicals. Studies on rodents undergoing repeated hepatic ischemia-reperfusion sessions suggest that hesperidin is a potential therapeutic agent for liver I/R lesions [39]. Finally, an important source of free radicals during infection are the enzyme NADPH oxidase of phagocytic cells (neutrophils, eosinophils, macrophages) which, activated by the inflammatory process, produce large quantities of toxic oxygen derivatives with a microbicidal function [46]. Under particular circumstances, when their production is in excess or the scavenger systems are inefficient or saturated, reactive oxygen species may escape from the cell that produces them and be released into the extracellular environment, becoming harmful and amplifying the injury due to inflammation [47,48].

Many studies have highlighted the importance of intracellular redox status as a new target for natural medicines, or synthetic drugs aimed at blocking both viral replication and virus-induced inflammation [38]. It has been suggested that, during COVID-19, the early treatment with antioxidants, such as *N*-acetylcysteine [49,50], melatonin [51,52], polyphenols [53,54,55], K, C, D, and E vitamins supplementation [56].

In this context, hesperidin might show some utility, due also to its antioxidant properties. Although no studies have been conducted so far directly aimed at proving hesperidin in COVID-19, the fact that it also has a powerful antioxidant action suggests that it could also have a beneficial effect through a protection mechanism against virus-induced cytotoxic damage.

Various in vitro and in vivo studies have shown that hesperidin’s antioxidant activity is not limited to its free radical scavenger activity, but also increased cellular defenses against oxidative stress and reduced inflammation makers via the ERK/Nrf2 signaling pathway [57]. Cisplatin-treated HK-2cells undergo oxidative stress and apoptosis, which are attenuated by hesperetin, by reducing ROS levels and activating the Nrf2 signaling pathway, which in turn regulates the antioxidant response elements [58]. Paracetamol is a common antipyretic and analgesic drug, but its overdose can cause acute liver failure, with a mechanism involving oxidative stress [59], that is mitigated by pre-treatment with hesperetin in a dose-dependent manner [60].

Hesperidin could be particularly useful in elderly people who suffer from greater oxidative stress. Although the reasons for a difference in severity of COVID-19 disease in subjects of different ages are unclear, it has been suggested that a key factor is the high antioxidant capacity of children, and the redox imbalance of elderly subjects with low antioxidant capacity [61,62], perhaps because the intracellular redox environment alters the presentation of antigens [63] and the expression or function of ACE2 [64,65]. The results of another study indicate the beneficial effects of citrus flavanones in the old-aged rat liver, where naringenin and hesperidin prevented the age-linked decrease of catalase, superoxide dismutase and glutathione reductase [66]. Hesperidin demonstrated antioxidant activity in rats after an intensive training program, and attenuated the secretion of cytokines by stimulated macrophages [67,68]. The administration of hesperetin has been shown to significantly reduce the levels of myeloperoxidase, malondialdehyde (a marker of lipid peroxidation) and inflammation in experimental models of colitis [69] and hepatic trauma [70]. In a model of rheumatic arthritis induced by Freund’s complete adjuvant, hesperidin successfully reversed the signs and symptoms, inflammatory markers and lipid peroxidation [71].

An interesting study compared the antioxidant capacity of the plasma of human subjects after the ingestion of 150 mL of different fruit juices [72]. A significant free radical elimination effect was observed already after 30 min, and up to 90 min after the ingestion of apple, orange, grape, peach, plum, kiwi, melon and watermelon juices, but not of pear juice. The grape juice showed a slightly longer lasting effect (up to 120 min after ingestion). No study, however, has evaluated the anti-viral effect of different fruits, but the data reported above in Section 2 and Section 3 suggest that it is mainly attributable to citrus fruits, due to their distinctive content of hesperidin.

## 5. Vitamin C

This article focuses more on hesperidin for its newly suggested anti-SARS-CoV-2 properties, but the importance of vitamin C, perhaps the best-known component of citrus fruits, cannot be overlooked. Vitamin C is the main antioxidant component of orange, and in normal nutrition it contributes, according to various authors, from 15% to 30% of the total antioxidant power of plasma [73]. The level of ascorbic acid in commercial orange juices (100%) ranges from about 35 mg/100 mL to about 74 mg/100 mL [73]. The consumption of blood oranges contributes to a daily intake of 9.4 mg/d (up to 55 mg/d) of anthocyanins and 58.5 mg/d (up to 340 mg/d) of vitamin C, respectively [74]. Citrus fruit samples (Sanguinello and Tarocco cultivars) showed vitamin C values higher than 54.9 mg/100 g of edible portion [75].

In COVID-19, a complementary therapeutic effect of intravenous high doses of vitamin C has been reported [76,77] and clinical trials are ongoing [78], but high doses of ascorbate may also be detrimental [79]. The role of dietary interventions is much more difficult to assess and any suggestion at present is just speculative or, at best, a working hypothesis [80].

There are conflicting data on the effect of vitamin C to prevent the common cold and other respiratory diseases. Coronaviruses are among the viruses that cause the common cold, a disease that has never had an effective cure or vaccine. Considering that SARS-CoV-2 is a coronavirus, and taking into account the low cost and high safety of natural foods rich in vitamin C, it has been suggested that it might be useful to increase the daily intake of these foods during the COVID-19 pandemic [80,81,82]. However, while many studies on the efficacy of vitamin C mega doses in preventing respiratory diseases are inconclusive or negative, meta-analyses suggest a consistent and statistically significant benefit of vitamin C for preventing the common cold and in people exposed to short periods of stress, intense exercise or in a cold environment [83,84].

Vitamin C, in addition to participating in the synthesis of collagen in the connective tissue, has a strong antioxidant effect, able to reduce the effects of free radicals, together with other vitamins, enzymes and minerals (zinc, selenium). Vitamin C is believed to prevent the oxidation of LDL and to protect human vascular smooth muscle cells from apoptosis [23] and boosts immune functions CARR2017}. Studies on animals infected with the flu virus have shown that vitamin C stimulates anti-viral immune responses and reduces the lungs’ inflammatory state [85,86]. We suggest that a beneficial effect of low-medium doses of vitamin C in the first stages of COVID-19 infection could also be due to the protection of cells from damage caused by the virus and/or by free radicals produced in the course of dysregulated inflammatory and immunopathological reactions.

The beneficial effects of an adequate amount of citrus fruits or of the integration of diet with vegetal extracts may result from the synergistic effects of their components [87], which provide protection against virus replication and oxidative damage.

## 6. Other Useful Effects

Hesperidin has multiple antimicrobial, antioxidant, anti-tumor, antihypertensive and immunostimulant medicinal properties [14,57,88,89,90,91,92,93,94,95,96,97]. Therefore, citrus fruits could have positive effects in the course of COVID-19 with additional mechanisms, besides the inhibition of virus replication and antioxidant activity.

In the most advanced stages, this disease presents multiple and complex systemic features: hypercoagulation, hyperactivation of the systemic inflammatory reactions, and a pathology that involves the blood vessels of the lung and other organs. For example, it has been argued that the mixture of hesperidin with diosmin co-administered with heparin protects against venous thromboembolism, which is a serious lung complication of the COVID-19 disease [98]. A randomized, single-blind, placebo-controlled, cross-over study in subjects with increased cardiovascular risk (aged 27 to 56 years) tested the administration of 500 mL of blood orange (dark red-colored *Citrus sinensis*) juice/day (or 500 mL of placebo/day) for periods of 7 days [99]. Endothelial function, measured as flow-mediated dilation, improved greatly and was normalized (5.7% compared to 7.9%; *p* < 0.005), after 1 week of consuming red orange juice. The concentrations of C-reactive protein, IL-6 and TNF-alpha also decreased significantly (*p* < 0.001).

Furthermore, this infection is known to affect elderly people with other cardiovascular and respiratory systems ailments. Consequently, any lifestyle-related intervention, including dietary interventions that increase hesperidin bioavailability [95] and help to maintain the health of the cardiovascular and respiratory systems, may make the person infected with SARS-CoV-2 less susceptible to its more severe complications. The risk of some chronic diseases like cerebrovascular disease and asthma is lower at higher dietary hesperetin intake [100], and a number of papers report beneficial effects in animal models of neurodegenerative disorders [96,101,102] and hyperthermia-induced febrile seizures [103]. Gene expression analysis has shown that hesperidin modulates the expression of genes involved in atherogenesis, inflammation, cell adhesion and cytoskeletal organization [104]. Physiologically relevant concentrations of flavanone reduce the adhesion of monocytes to endothelial cells stimulated by TNF-alpha, influencing the expression of related genes, and offering a potential explanation of its vasculoprotective effects. A daily dose of 292 mg of hesperidin, corresponding to 500 mL of orange juice, was sufficient to achieve the described effects.

A randomized controlled crossover study [105] of 24 healthy and overweight men (age 50–65 years) investigated the effects of orange and hesperidin on the vascular system. During three periods of four weeks, the volunteers consumed 500 mL of orange juice, 500 mL of control drink, plus hesperidin, or 500 mL of control drink plus placebo every day. After 4 weeks of consuming orange juice or control drink plus hesperidin, the diastolic pressure had significantly decreased compared to control drink plus placebo (*p* = 0.02). Both orange juice and control drink plus hesperidin ingestion improved postprandial microvascular endothelial reactivity compared to placebo (*p* < 0.05), measured at the peak of plasma concentration of hesperetin. The authors conclude that, in healthy middle-aged men in moderate overweight, the regular consumption of orange juice reduces diastolic pressure and increases endothelium-dependent microvascular reactivity. The study suggests that this beneficial effect is due to hesperidin. Various in vivo experiments revealed the protective effects of hesperidin against the inflammation produced by lipopolysaccharide (LPS) in liver and spleen [106]. 

Studies in mice showed protective effects of hesperetin in LPS-induced neuroinflammation, neuronal oxidative stress and memory impairment [107]. Hesperetin significantly reduced the expression of inflammatory cytokines in microglia, and attenuated the generation of reactive oxygen species induced by LPS. In addition, hesperetin improved synaptic integrity, cognition and memory processes. In a recent review, it was noted that the nutraceutical, antioxidant and anti-inflammatory properties of hesperidin could be useful also in neurodegenerative diseases [101]. A dietary medical history study determined the total dietary intake of 10,054 Finnish men and women in the previous year [100]. Flavonoids intake in food was estimated and compared with the incidence of diseases considered by different national public health registers. People with higher hesperetin intakes had lower incidences of cerebrovascular disease (RR 0.80; CI 0.64–0.99; *p* = 0.008) and bronchial asthma (RR 0.64; CI 0.46–0.88; *p* = 0.03).

The effect of orange juice on inflammation and oxidative stress induced by a high-fat meal was studied [108] in three groups of 10 normal and healthy subjects, invited to drink water, or 300 kcal of glucose, or juice orange, in combination with a high-fat meal of 900 kcal. In blood samples obtained before, and 1, 3 and 5 h after the consumption of the meal and of the different drinks, some indexes of inflammation were determined. The high-fat meal increased the expression of NADPH oxidase, toll-like receptors and metalloproteinase-9 in mononuclear cells and plasma. These changes were significantly reduced by the intake of orange juice. Other authors [72,109] have also described antioxidant effects in healthy volunteers after taking orange juice and whole fruits rich in vitamin C.

Since COVID-19 disease is a multi-organ disease and has more serious clinical consequences in subjects suffering from co-morbidities and cardiovascular pathologies, it is conceivable that its clinical course could benefit from the multiple valuable effects of hesperidin in systemic and chronic-degenerative pathologies.

## 7. Discussion

The scientific literature on the healthy properties of fruit and vegetables [5,18,29,110,111,112,113,114] is vast and beyond the scope of this article, which has focused on the remarkable and surprising interaction between hesperidin, and the key proteins of the SARS-CoV-2 virus, seen by means of computational simulations. Since these methods are now the “gold standard” for screening new drugs and their targets, we can hypothesize the beneficial effects of hesperidin in COVID-19 as well, pending the need of clinical evidence of therapeutic efficacy. The binding of hesperidin to the central part of the spike and to the main protease is much stronger than that of conventional antivirals, and it can be expected that this molecule may soon be tested in randomized trials of patients with COVID-19 or subjects exposed to contagion, as is the case for quercetin, or for a mix of quercetin, green tea, cinnamon and liquorice [115]. These new pharmacological properties of hesperidin are added to those of an antioxidant agent, already known.

A systematic review with dose-response meta-analysis of prospective studies studying the relationship between fruit, vegetables and cardiovascular diseases, total cancer and all-cause mortality [116], found a non-linear relationship between citrus fruits intake and all-cause mortality, with the nadir for an average consumption between 50 and 100 g/day, and with an apparent tendency of the dose-response curve to lose any benefit around 250 g per day. This might happen because of sweetened juices, as the consumption of sugary drinks/high-sugar beverages in the USA was associated with an increase in mortality [117]. An average consumption of citrus fruit close to 100 g/day would seem optimal, preferring whole fruits, which are associated with a maximum intake also of hesperidin, in addition to dietary fiber and other nutrients, in comparison with juices or centrifuged drinks.

In general, it cannot be argued that citrus fruits are healthier than other fruits [72,116], but it is certainly true for their hesperidin content, and therefore for their possible protective effect against COVID-19. The recently accumulated evidence suggests that hesperidin supplementation may be useful as prophylactic agent against SARS-CoV-2 infection and as complementary treatment during COVID-disease, as recently suggested by others [24,98]. In support of this hypothesis, the latter work cited the in silico study of Wu et al. [31], and proposed that the benefit of hesperidin may derive both from the binding to the coronavirus spike and from its anti-inflammatory activity. As it appears from our review, several other research groups [32,33,34,35,36] have been added to the work of Wu et al. [31], showing low binding energy to other viral proteins besides to spike. In this work, we have also given importance to further biological actions of citrus fruits, in protecting the cell from damage caused by the virus and the oxidative stress. Moreover, from our point of view, it is important to consider, in addition to the effect of the single molecule, also the set of benefits of citrus fruits and whole fruit juice. In fact, orange, lemon and mandarin contain a significant amount of hesperidin that can be taken through the diet, and contain also vitamin C, which has nutraceutical properties that could synergize with the flavanone.

Whether regular citrus consumption, or an increase in consumption, may be advisable among the preventive dietary measures for COVID-19 is a matter of future investigations. A dose of citrus fruits or vitamin C-based supplements, higher than that of a typical diet of the Italian population, does not seem suitable for long-term prevention. However, in periods of intense stress (that may be considered similar to the exposure to pathogenic microorganisms during the epidemic peak or during another infectious disease), possible benefits from high doses of vitamin C are expected [84]. It was suggested that the prevention of infection requires dietary intakes of vitamin C (i.e., 100–200 mg/day) [118], which provide adequate plasma levels to optimize cell and tissue levels, while the treatment of established infections may probably require significantly higher doses; high (grams) in vitamin to compensate for the increased inflammatory response and metabolic demand [119]. If you rely on food, these doses may be reached with a temporary large consumption of juices, taking care to crush well also the albedo, which is the part richest in hesperidin.

Following the computational evidence of the interaction between hesperidin and key viral proteins, it is likely that this component will become part of the candidate drugs for a preventative or therapeutic effect. In order to support a similar effect of dietary doses, adequate epidemiological studies would be needed to compare the incidence of COVID-19 in populations with different dietary intake of oranges and other citrus fruits, in an analogy with studies showing statistically significant benefits of specific diet components in infectious diseases [1,120] and in cancer of the digestive tract [121]. We reported on the bioavailability of hesperidin (about 2 micromol/L in plasma after ingestion of 500 mL of juice [25]) and the cited article, showing that micromolar doses of hesperidin inhibit the main protease of the SARS virus [37]; this suggests that an infection-blocking effect could be approached or achieved, even with an increase in citrus fruits intake for a certain period, particularly when consumed with peel and albedo, richer in hesperidin than juice [24]. Furthermore, it is conceivable that a high dose of nutraceutical principles should be present during and after the intake in the oral cavity and in the digestive tract, thus providing a local impediment to virus entry and replication in these anatomic sites, which play a crucial role in the COVID-19 disease [9,122,123].

Finally, quarantine is likely to have also negative effects on lifestyle and, including those related to stress, anxiety, reduced physical activity and nutrition, with a consequent low intake of antioxidants and vitamins. It has been suggested that, during quarantine, strategies to encourage adherence to an adequate diet rich in fruits and vegetables should be implemented [124].

## 8. Conclusions

In conclusion, what we have reported here elucidates the multiple biological actions of hesperidin and vitamin C, two major components of citrus fruits which appear to be effective candidates to counteract the cell infection by SARS-CoV-2, and to modulate the systemic immunopathological phases of the disease. Further preclinical, epidemiological and clinical studies are needed to corroborate the hypothesis that an adequate intake of citrus fruits or their extracts could effectively contribute to the strategies for the prevention of COVID-19.

## Figures and Tables

**Figure 1 antioxidants-09-00742-f001:**
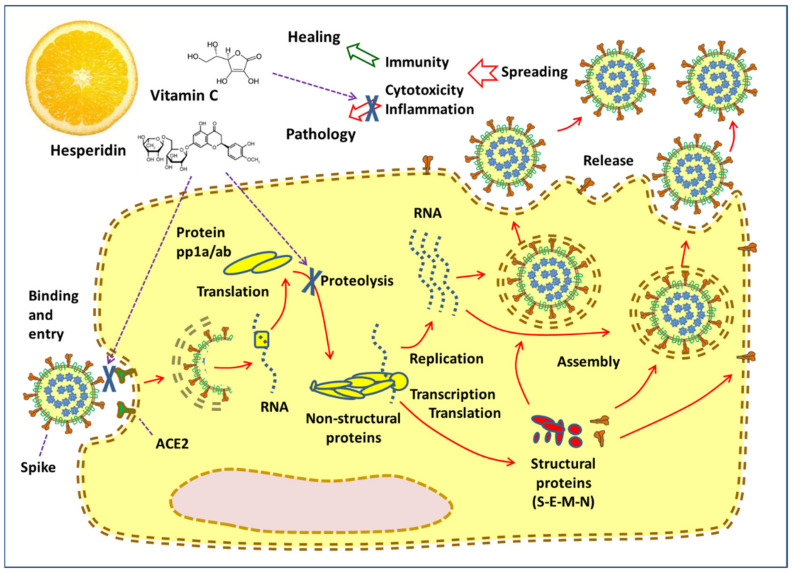
Cellular cycle of the Severe acute respiratory syndrome coronavirus 2 (SARS-CoV-2) virus and the places of the inhibition of virus-induced cellular and systemic pathology by hesperidin and vitamin C (indicated with “X”).

**Figure 2 antioxidants-09-00742-f002:**
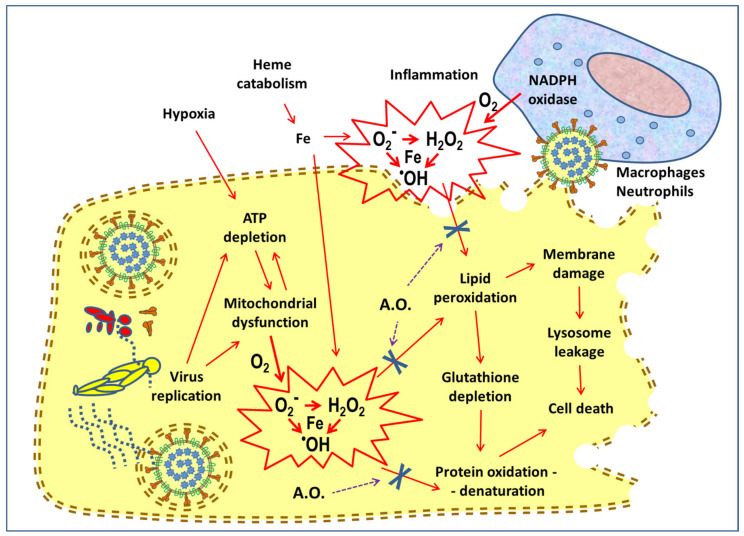
Schematic representation of the mechanisms of generation of oxygen free radicals in the course of Coronavirus disease 2019 (COVID-19), and assumptions about the antioxidant (A.O.) action sites, indicated with “X”.

**Table 1 antioxidants-09-00742-t001:** Hesperidin content (mg/100 mL of fresh juice) in different citrus fruits. Data are from the reviews of Gattuso et al. [22] and, for red orange, of Grosso et al. [23]

Fruit	Hesperidin Content (mg/100 mL Juice)			
	mg	SD	Min.	Max.
*Citrus sinensis* (sweet orange)	28.6	11.9	3.5	55.2
*Citrus sinensis* (red orange)	43.6	17.9	18	66.5
Commercial sweet orange juice	37.5	19.2	4.45	76.3
*Citrus reticulata* (mandarin)	24.3	18.2	0.81	45.8
*Citrus clementine* (clementine)	39.9	29.4	5.21	86.1
*Citrus limon* (lemon)	20.5	12.4	3.84	41
*Citrus aurantifolia* (lime)	1.8	0.35	1.52	2.0
*Citrus paradisi* (grapefruit)	0.9	0.58	0.25	1.8
Commercial grapefruit juice	2.8	3.9	0.2	16.4

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
