# Peer review of "Hesperidin and SARS-CoV-2: New Light on the Healthy Function of Citrus Fruits"

_antioxidants, 2020, doi:10.3390/antiox9080742_

Round 1
Reviewer 1 Report
Page 1, line 11, Rewrite the sentence “…,possible…marginal”
Page 1, line 12, “Nutrition” has to be changed in “Foods”
Page 1, line 16, Delete the sentence “well known for its vitamin C content, but less for the function of its flavonoids.”
Page 1, line 18, delete the sentence “well known for its vitamin C content, but less for the function of its flavonoids.”
Page 1, line 22, “…is comparable if not superior to…” what does it mean? the sentence may be may be more precise
Page 1, line 22, delete the word “chemical” and mention some antivirals.
Page 1, line 26, “increased pressure on the immune system” has to be rewritten more clearly.
Page 1, line 26, “the reasons…discussed.” has to be rewritten
Page 1, line 32, change the sentence “COVID-19…interventions.” I dont’t think COVID-19 gave an opportunity. They were in difficult situations.
Page 1, line 43, change in this way “….bioactive substances such as polyphenols and vitamins including…”
Page 2, line 49, the sentence has to be rewritten.
Page 2, line 53, “Citrus….C” has to be rewritten
Page 2 , line 57, delete “There are indications that”
Page 2, line 58, “virus” has to be chancged in “viral”
page 2, line 60, do you hypothize that Orange mey be useful for hesperidin?
Page 2, line 72, What do you mean with “introduction”. You may better specify. Is it bioavailability? or something else? In addition add a reference.
Page 2, line 82, “Some tests have tried to assess…” they have demonstrated or not. Correct the sentence
Page 2, line 83, rewrite the sentence “Healthy…kinetics.”
Page 2, line 86, correct “Elimination half-lives” Halflife is referred to one compound only.
Page 3, line 116, correct the sentence
Page 3, line 124, correct “filament”
Page 4, line 149, correct “augmentation of the viral…”
Page 4, line 152, delete “if not blocked”
Page 4, line 155, correct “as antagonists of cytokines or TNF”. Is not TNF a cytokine???
Page 4, lines 158-163 should be put in another paragrgaph concenring safety data.
Page 4, line 167, correct “to try to verify”
Page 4, line 169, delete “as it is performed through the computer (with reference 169 to the silicon of the chips).”
Page 4, line 173, correct “tested”
Page 5, lines 194-196, rewrite the sentence.
Page 5, line 199, rewrite the sentence
Page 5, line 213, rewrite the sentence.
Page 5, lines 216-218. Delete the sentence
Page 5, line 218, correct “it is suggestive”
Page 5, line 225-226, rewrite the sentence.
Page 6, line 228, change the word “powerfull”.
Page 6, line 240, correct “classic mechanisms” in “experimental model” and rewrite the sentence.
page 6, line 251, find another word for “quantities”
Page 6, line 256, corrects “vitamine”
page 7, line 258, correct “prove” and delete “which it has in common with other polyphenols”
Page 7, line 261, “cytoprotection”
Page 7, lines 267-271, “Paracetamol…stress.” rewrite the whole concept sentence
Page 7, line 272, delete “are known to”
Page 7, line 290, delete “can”.
Page 7, line 291, correct “and not only” and rewrite the sentence in a more appropriate English.
Page 7, line 293, delete “especially if in a severe form” and write “cariovascular and respiratory systems health”
Page 7, line 303, what’s “blood orange juice / day” ?
Page 8, line 311, correct “three 4-week periods”
Page 8, line 320, delete “on laboratory mice” and “have”
Page 8, line 321, correct “produced by the negative effects of lipopolysaccharide (LPS)”
Page 8, line 324, correct “Intakes of flavonoids in food” in “flavonoids intake”
Page 8, line 328, correct “In mouse studies”
Page 8, lines 329-333, “While…LPS.” rewrite the sentence in a more concise way.
Page 8, line 336, rewrite “disorders of the central nervous system”
Page 8, line 339, delete “so”
Page 8, line 345, rewrite “will also indirectly profit”
Page 9, line 365, rewrite “whether…pandemic”
Page 9, line 376, correct “inflammatory state of the lung” in “lungs inflammatory state”
Page 9, line 378, rewrite “out-of-control”
Page 9, line 382, correct “On in the blood”
Page 9, line 386, delete “indicative of oxidative stress”
Page 9, line 389, correct “can depend on” in “may result from”
page 9, lines 390-392, rewrite the concept. If these substances may be useful in COVID-19, they shoud administered within vegetal extracts and not in foods.
Page 9, line 397, change the sentence “we feel hopeful” in “we hypotize” and change the sentence
page 9, lines 404-404, rewrite the sentence
page 10, line 416, Correct “It remains to be seen”. probably it is the translation of the italian “rimane da vedere”, however in English it is wrong.
Page 10, line 421, correct “therapeutic” in “high”.
Page 10, line 424, correct “require” in “may probably require”
Page 10, lines 427-429, delete “In our opinion…century!”
Page 10, line 430, delete “strong and consistent”
Page 10, line 431, change “it is easy to imagine”. It is a scientific article.
Page 10, line 432, “preventative protection can be obtained with food doses” is not in agreement with previous hypotesis reported in the manuscript.
Page 10, lines 432-440. this hypotesis is wrong. rewrite the concept.
Page 10, line 441, add the word “also” before “negative”
Page 10, lines 445-452, rewrite the sentence.
“The orange is the fruit of various Citrus species of the Rutaceae family.” has to be rewritten in another way as it is identical in the following paper.
https://www.google.com/url?sa=t&rct=j&q=&esrc=s&source=web&cd=&cad=rja&uact=8&ved=2ahUKEwixw9_cyfnqAhUJCuwKHagbD28QFjAAegQIARAB&url=https%3A%2F%2Fwww.preprints.org%2Fmanuscript%2F202006.0321%2Fv1%2Fdownload&usg=AOvVaw2g5YYeH4V_pCIla7X5puzv
“
Vitamin C represents the highest fraction of total antioxidant activity of orange juice, 350 contributing, according to various studies, for 60% up to 87% and being in charge of…” has to be rewritten as it is identical to the following paper
https://www.ncbi.nlm.nih.gov/pmc/articles/PMC6123350/
Author Response
Response to Reviewer 1 Comments
We sincerely thank the Reviewer for his/her careful notes, which were very useful to improve the paper.
Page 1, line 11, Rewrite the sentence “…,possible…marginal”
Response: OK, we have rewritten the sentence as follows: “Among the many approaches to COVID-19 prevention, the possible role of nutrition has so far been rather underestimated”.
Page 1, line 12, “Nutrition” has to be changed in “Foods”
R: OK, done as suggested
Page 1, line 16, Delete the sentence “well known for its vitamin C content, but less for the function of its flavonoids.”
R: To make the sentence more correct and to preserve the meaning of the paragraph, we have rewitten the sentence as follows: “well known for its vitamin and flavonoid content”.
Page 1, line 22, “…is comparable if not superior to…” what does it mean? the sentence may be may be more precise
R: OK. Thanks to this right suggestion and the following one, we reformulated the sentence as follows: “The binding energy of hesperidin to these important components is lower than that of lopinavir, ritonavir, indinavir, suggesting that it could perform an effective antiviral action.”
Page 1, line 22, delete the word “chemical” and mention some antivirals.
R: OK, done as suggested
Page 1, line 26, “increased pressure on the immune system” has to be rewritten more clearly.
R: OK, we have rewritten the sentence as follows: “There is discussion about the preventive efficacy of vitamin C, at the dose achievable by the diet, but recent reviews suggest that this substance can be useful in case of strong immune system burden caused by viral disease.”
Page 1, line 26, “the reasons…discussed.” has to be rewritten
R: OK, we have rewritten the sentence as follows: “Computational methods and laboratory studies support the need to undertake apposite preclinical, epidemiological, and experimental studies on the potential benefits of Citrus fruit components for the prevention of infectious diseases, including COVID-19.”
Page 1, line 32, change the sentence “COVID-19…interventions.” I dont’t think COVID-19 gave an opportunity. They were in difficult situations.
R: We only partially agree with the Reviewer, because in reality the disease gave the opportunity to try various treatments, precisely because there was no standard therapy. But we understand that the word "opportunity" can be misinterpreted and therefore we have changed the sentence as follows: “COVID-19, being a new and largely unknown disease, has put doctors in the need to investigate and try new approaches and interventions”
Page 1, line 43, change in this way “….bioactive substances such as polyphenols and vitamins including…”
R: OK, done as suggested
Page 2, line 49, the sentence has to be rewritten.
R: OK, we have rewritten as follows: “This article considers the nutraceutical properties of Citrus fruits, with particular attention to hesperidin and vitamin C as potential anti-SARS-CoV-2 activities, as antioxidants, and as modulators of inflammation.”
Page 2, line 53, “Citrus….C” has to be rewritten
R: OK, we have rewritten as follows: “Citrus fruits are rich sources of vitamin C, anthocyanins, and flavanones, with hesperidin...
Page 2 , line 57, delete “There are indications that”
R: OK, done as suggested
Page 2, line 58, “virus” has to be chancged in “viral”
R: OK, done as suggested
page 2, line 60, do you hypothize that Orange mey be useful for hesperidin?
R: The evidence from several computational studies allows us to formulate the hypothesis that the most "specific" antiviral action towards SARS-CoV-2 is provided by hesperidin. In various parts of the work, the role of vitamin C is clarified, which however seems to be more non-specific, that is, antioxidant and supporting the cell's vitality.
Page 2, line 72, What do you mean with “introduction”. You may better specify. Is it bioavailability? or something else? In addition add a reference.
R: We agree that the sentence is unclear and we have rewritten it as follows, including a reference: “The latter is present mainly in the peel and in the white part (albedo) of Citrus fruits, and the consumption of the whole fruits may allow a greater intake than the juice [21]. In fact, in fresh orange juice the content of hesperidin is about 30 mg per 100 ml and in commercial juice it can be a little higher [22], probably because the industrial processing incorporates more peel.”
Page 2, line 82, “Some tests have tried to assess…” they have demonstrated or not. Correct the sentence
- OK we have corrected in: “Some tests have assessed the amount of hesperidin (or its metabolite hesperetin) in the blood of people drinking orange juice”
Page 2, line 83, rewrite the sentence “Healthy…kinetics.”
- OK we have rewritten in: “Healthy volunteers drank orange juice in one intake (8 ml / kg) and blood and urine samples were collected between 0 and 24 hours after administration [25]. The peak plasma concentration of hesperetin was 2.2 ±1.6 micromol/L, with significant variations in different subjects. Elimination half-life ranged from 1.3 to 2.2 hours, indicating short-term kinetics.”
Page 2, line 86, correct “Elimination half-lives” Halflife is referred to one compound only.
R: OK, done as suggested
Page 3, line 116, correct the sentence
- OK we have rewritten as follows: “The internalisation of SARS-CoV-2 is mediated by the binding of the spike glycoprotein of the virus with its receptor (ACE2) on the cell membranes. ACE2 is expressed in several tissues, including alveolar lung cells, gastrointestinal tissue and even the brain [7-10].”
Page 3, line 124, correct “filament”
R: OK we have rewritten as follows: “To this purpose, structural proteins are incorporated into the membrane and the nucleocapsid N protein combines with the positive-sense RNA produced through the replication process, to become a nucleoprotein complex.”
Page 4, line 149, correct “augmentation of the viral…”
R: OK we have rewritten as follows: “...this mechanism finds a complication in the risk of enhancement of the viral entry into target cells by the same antibodies ("antibody-dependent-enhancement") [12] or autoimmune reactions [11].”
Page 4, line 152, delete “if not blocked”
R: OK, done as suggested
Page 4, line 155, correct “as antagonists of cytokines or TNF”. Is not TNF a cytokine???
R: OK, we have rewritten the sentence as follows: “with new biological agents such as receptor antagonists or anti-cytokine antibodies.”
Page 4, lines 158-163 should be put in another paragrgaph concenring safety data.
R: OK, we have put the safety data at the end of the paragraph n. 2 on the bioavailability.
Page 4, line 167, correct “to try to verify”
R: OK, we have rewritten the sentence as follows: “The researchers started from the detailed knowledge of the virus protein structure to ascertain which molecules, natural or artificial, are capable of binding...”
Page 4, line 169, delete “as it is performed through the computer (with reference 169 to the silicon of the chips).”
R: OK, we have rewritten the sentence as follows: “This technique, called "in silico", is currently applied to predict drug behavior and accelerate the detection rate...”
Page 4, line 173, correct “tested”
R: OK, we have rewritten the sentence as follows: “... have tested 1066 natural substances with potential antiviral effect, plus 78 antiviral drugs already known in the literature, for their binding to SARS-CoV-2 proteins”
Page 5, lines 194-196, rewrite the sentence.
R: OK, we have rewritten the sentence as follows: “Hesperidin binds with hydrogen bonds to various amino acids, mainly THR24, THR25, THR45, HIS4, SER46, CYS145.”
Page 5, line 199, rewrite the sentence
R: OK, we have rewritten the sentence as follows: “A research published by Indonesian authors and so far available in Preprints has examined with computational methods a wide range of active principles of the medicinal plants Curcuma sp., Citrus sp. (orange), Caesalpinia sappan and Alpinia galanga for their ability of "molecular docking" towards viral proteins”
Page 5, line 213, rewrite the sentence.
R: OK, we have rewritten the sentence as follows: “The flavanone interacts with several amino acids of the protein through hydrogen bonds and the interaction of hesperidin is more effective than that realised by the reference drug nelfinavir (with scores of -178.59 and -147.38 respectively).”
Page 5, lines 216-218. Delete the sentence
R: OK, done as suggested
Page 5, line 218, correct “it is suggestive”
R: OK, we have rewritten the sentence as follows: “Interestingly, this research suggests that the inhibition of viral protease occurs at concentrations of hesperidin of the same order of magnitude as those achievable in plasma with a large oral supplement of orange juice.”
Page 5, line 225-226, rewrite the sentence.
R: OK, we have rewritten the sentence as follows:“An efficient oxidative metabolism at the mitochondrial level (without unwarranted formation of free radicals) and the balance of oxidation reactions, due to the intervention of the enzymatic systems and various scavenger molecules, are essential for the vitality of the cells of each tissue”
Page 6, line 228, change the word “powerfull”.
Response: OK, the sentence has been changed as follows: “Hesperidin contributes significantly to antioxidant defense systems as an effective agent against superoxide and hydroxyl radicals [39], and its derivative hesperetin inhibits nitric oxide production by LPS-stimulated microglial cells [40].”
Page 6, line 240, correct “classic mechanisms” in “experimental model” and rewrite the sentence.
R: OK, we have rewritten the sentence as follows: “Another experimental model of generation of free radicals is represented by ischemia and reperfusion (I/R). This occurrence has been described also in COVID-19 [44] or can aggravate the treatment with drugs such as chloroquine [45]: during prolonged hypoxia of a tissue, the cells undergo structural damage, especially in the mitochondria and the endoplasmic reticulum.”
page 6, line 251, find another word for “quantities”
Response: OK, the word has been changed to “amounts”
Page 6, line 256, corrects “vitamine”
Response: OK, the word has been changed to “vitamins”
page 7, line 258, correct “prove” and delete “which it has in common with other polyphenols”
Response: OK, the sentence has been changed to : “In this context, hesperidin might show some utility due also to its antioxidant properties.”
Page 7, line 261, “cytoprotection”
Response: OK, the sentence has been changed to : “suggests that it could also have a beneficial effect through a protection mechanism against virus-induced cytotoxic damage”.
Page 7, lines 267-271, “Paracetamol…stress.” rewrite the whole concept sentence
Response: OK, the sentence has been rewritten as: “Paracetamol is a common antipyretic and analgesic drug, but its overdose can cause acute liver failure, with a mechanism involving oxidative stress [59], that is mitigated by pre-treatment with hesperetin in a dose-dependent manner [60].”
Page 7, line 272, delete “are known to”
Response: OK, done
Page 7, line 290, delete “can”.
Response: OK, done
Page 7, line 291, correct “and not only” and rewrite the sentence in a more appropriate English.
Response: OK, the sentence has been rewritten as: “In the most advanced stages, this disease presents multiple and complex systemic features: hypercoagulation, hyperactivation of the systemic inflammatory reactions, and a pathology that involves the blood vessels of the lung and other organs.”
Page 7, line 293, delete “especially if in a severe form” and write “cariovascular and respiratory systems health”
Response: OK, the sentence has been rewritten as: “Furthermore, this infection is known to affect elderly people with other cardiovascular and respiratory systems ailments.”
Page 7, line 303, what’s “blood orange juice / day” ?
Response: The blood orange is a red variety of orange, with almost blood-colored pulp and juice. The distinctive dark color is due to the presence of anthocyanins. We have addes a short explanation “(dark red-colored Citrus sinensis) “
Page 8, line 311, correct “three 4-week periods”
Response: OK, the sentence has been rewritten as: “During three periods of 4 weeks”
Page 8, line 320, delete “on laboratory mice” and “have”
Response: OK, done
Page 8, line 321, correct “produced by the negative effects of lipopolysaccharide (LPS)”
Response: OK, the sentence has been rewritten as: “Various in vivo experiments revealed the protective effects of hesperidin against the inflammation produced by lipopolysaccharide (LPS) in liver and spleen”
Page 8, line 324, correct “Intakes of flavonoids in food” in “flavonoids intake”
Response: OK, done
Page 8, line 328, correct “In mouse studies”
Response: OK, the sentence has been rewritten as: “Studies in mice showed protective effects of hesperetin in LPS-induced neuroinflammation, neuronal oxidative stress and memory impairment”
Page 8, lines 329-333, “While…LPS.” rewrite the sentence in a more concise way.
Response: OK, the sentence has been rewritten as: “Hesperetin significantly reduced the expression of inflammatory cytokines in microglia and attenuated the generation of reactive oxygen species induced by LPS.”
Page 8, line 336, rewrite “disorders of the central nervous system”
Response: OK, the sentence has been rewritten as: “In a recent review, it was noted that the nutraceutical, antioxidant and anti-inflammatory properties of hesperidin could be useful also in neurodegenerative diseases”
Page 8, line 339, delete “so”
Response: OK, done
Page 8, line 345, rewrite “will also indirectly profit”
Response: OK, the sentence has been rewritten as: “it is conceivable that its clinical course could benefit from the multiple valuable effects of hesperidin in systemic and chronic-degenerative pathologies.”
Page 9, line 365, rewrite “whether…pandemic”
Response: OK, the sentence has been rewritten as: “it has been suggested that it might be useful to increase the daily intake of these foods during the COVID-19 pandemic [80-82].”
Page 9, line 376, correct “inflammatory state of the lung” in “lungs inflammatory state”
Response: OK, done
Page 9, line 378, rewrite “out-of-control”
Response: OK, the sentence has been rewritten more clearly as: “We suggest that a beneficial effect of low-medium doses of vitamin C in the first stages of COVID-19 infection could also be due to the protection of cells from damage caused by the virus and/or by free radicals produced in the course of dysregulated inflammatory and immunopathological reactions.”
Page 9, line 382, correct “On in the blood”
Response: OK, done
Page 9, line 386, delete “indicative of oxidative stress”
Response: OK, done
Page 9, line 389, correct “can depend on” in “may result from”
Response: OK, done
page 9, lines 390-392, rewrite the concept. If these substances may be useful in COVID-19, they shoud administered within vegetal extracts and not in foods.
Response: The sentence concerned the important point of the synergistic effect of Vitamin C with other vegetal compounds, and following the advice of the Reviewer has been rewritten as: “The beneficial effects of an adequate amount of Citrus fruits or of integration of diet with vegetal extracts may result from the synergistic effects of their components [87], which provide protection against virus replication and oxidative damage..”
Page 9, line 397, change the sentence “we feel hopeful” in “we hypotize” and change the sentence
Response: OK, the sentence has been rewritten as: “Since these methods are now the "gold standard" for screening new drugs and their targets, we can hypotize the beneficial effects of hesperidin in COVID-19 as well, pending the need of clinical evidence of therapeutic efficacy.”
page 9, lines 404-404, rewrite the sentence
Response: OK, the sentence has been rewritten as: “These new pharmacological properties of hesperidin are added to those of an antioxidant agent, already known.”
page 10, line 416, Correct “It remains to be seen”. probably it is the translation of the italian “rimane da vedere”, however in English it is wrong.
Response: OK, the sentence has been rewritten as: “Whether regular Citrus consumption, or an increase in consumption, may be advisable among the preventive dietary measures for COVID-19 is a matter of future investigations.”
Page 10, line 421, correct “therapeutic” in “high”.
Response: OK, done
Page 10, line 424, correct “require” in “may probably require”
Response: OK, done
Page 10, lines 427-429, delete “In our opinion…century!”
Response: OK, done
Page 10, line 430, delete “strong and consistent”
Response: OK, done.
Page 10, line 431, change “it is easy to imagine”. It is a scientific article.
Response: OK, the sentence was then changed to: “it is conceivable”
Page 10, line 432, “preventative protection can be obtained with food doses” is not in agreement with previous hypotesis reported in the manuscript.
Response: OK, the sentence was then changed to: “Following the computational evidence of the interaction between hesperidin and key viral proteins, it is conceivable that this component will become part of the candidate drugs for a preventative or therapeutic effect.”
Page 10, lines 432-440. this hypotesis is wrong. rewrite the concept.
Response: We understand the reviewer's comments on this point and share them only in part. In fact, it is certainly foreseeable that a concentrated plant extract or a high dose of a purified molecule may have sharper pharmacological effects than the intake of the same active ingredients with the diet. On the other hand, all our work focuses on the benefits of the whole fruit, which has an advantage both for the multiplicity of components and for the ease of taking in normal living conditions. Today, the nutraceutical properties of dietary vegetables are accepted in many fields of medicine, including infectious diseases. For example, we recall what systematic reviews show (with dose-response meta-analysis): a consumption of at least 50 g / day of whole grains was associated with a 20% reduction in mortality from infectious diseases, which reaches 26% in the dose-analysis {AUNE2016A}; a consumption of one portion per day of oily dried fruit (total nuts) was associated with a 73-75% reduction in mortality from infections {AUNE2016}; for every additional 200 g per day of fruit vegetables was associated with a 10% more reduction in total mortality (RR 0.90; 0.87-0.93), which probably also includes that from infection, affecting all countries of the world (Supplementary material to Aune D, ref. 97 {AUNE2017}) , including those where infectious diseases are important determinants of total mortality. Furthermore, it is important to underline that the components taken with the diet have their particular efficacy on the mucous membranes of the oral cavity and digestive tract, as has already been known from epidemiological studies in the field of neoplasms {ZANINI2015}. We therefore believe that our idea, presented in the form of a hypothesis, is legitimate and not wrong. Taking into account the auditor's note, to express more clearly our view, we have reformulated the entire paragraph as follows: “Following the computational evidence of the interaction between hesperidin and key viral proteins, likely this component will become part of the candidate drugs for a preventative or therapeutic effect. In order to support a similar effect of dietary doses, adequate epidemiological studies would be needed to compare the incidence of COVID-19 in populations with different dietary intake of oranges and other citrus fruits, in analogy with studies showing statistically significant benefits of specific diet components in infectious diseases [1, 120] and in cancer of digestive tract [121]. What reported on the bioavailability of hesperidin (about 2 micromol/L in plasma after ingestion of 500 ml of juice [25]) and the cited article showing that micromolar doses of hesperidin inhibit the main protease of the SARS virus [37], suggest that an infection-blocking effect could be approached or achieved even with an increase in Citrus fruits intake for a certain period, particularly when consumed with peel and albedo, richer in hesperidin than juice [24]. Furthermore, it is conceivable that a high dose of nutraceutical principles should be present during and after the intake in the oral cavity and in the digestive tract, thus providing a local impediment to virus entry and replication in these anatomic sites, which play a crucial role in the COVID-19 disease [9] [122] [123].”
Page 10, line 441, add the word “also” before “negative”
Response: OK, done
Page 10, lines 445-452, rewrite the sentence.
Response: OK, the sentence has been rewritten in a more synthetic way, omitting the unnecessary part of intervention studies: “In conclusion, what we have reported here elucidates the multiple biological actions of hesperidin and vitamin C, two major components of Citrus fruits which appear to be effective candidates to counteract the cell infection by SARS-CoV-2 and to modulate the systemic immunopathological phases of the disease. Further preclinical, epidemiological and clinical studies are needed to corroborate the hypothesis that an adequate intake of Citrus fruits or their extracts could effectively contribute to the strategies for the prevention of COVID-19.”
“The orange is the fruit of various Citrus species of the Rutaceae family.” has to be rewritten in another way as it is identical in the following paper. https://www.google.com/url?sa=t&rct=j&q=&esrc=s&source=web&cd=&cad=rja&uact=8&ved=2ahUKEwixw9_cyfnqAhUJCuwKHagbD28QFjAAegQIARAB&url=https%3A%2F%2Fwww.preprints.org%2Fmanuscript%2F202006.0321%2Fv1%2Fdownload&usg=AOvVaw2g5YYeH4V_pCIla7X5puzv
Response: OK, we have removed this sentence because is not necessary in the context of the revised paper.
“Vitamin C represents the highest fraction of total antioxidant activity of orange juice, contributing, according to various studies, for 60% up to 87% and being in charge of…” has to be rewritten as it is identical to the following paper https://www.ncbi.nlm.nih.gov/pmc/articles/PMC6123350/
Response: OK, the sentence (to which we added the correct citation) has been rewritten as: “Vitamin C is the main antioxidant component of orange and in normal nutrition it contributes, according to various authors, from 15% to 30% of the total antioxidant power of plasma [73].”
Reviewer 2 Report
The manuscript by Bellavite and Donzelli represents a review article focused on potential benefits of Citrus fruits on public health and their effects on infectious disease prevention. The article summarizes a set of interesting information. However, I would like to ask authors clarifying some details:
- The second chapter is describing the useful contents of Citrus fruits. It contains a lot of data on these contents and their concentrations. However, in my opinion, such presentation within the text makes comparisons analysis impossible. The table containing the information on the quantities of components in different Citrus fruits could be a solution;
- I am really interested to know the opinion of the authors what Citrus fruit (and in which form) is the best one;
- In my opinion short description of the benefits of Citrus fruits in comparison to other fruits should be also included.
Author Response
Response to Reviewer 2 Comments
- The manuscript by Bellavite and Donzelli represents a review article focused on potential benefits of Citrus fruits on public health and their effects on infectious disease prevention. The article summarizes a set of interesting information.
Response: We really appreciate this positive comment of the Reviewer to our workHowever, I would like to ask authors clarifying some details:
- The second chapter is describing the useful contents of Citrus fruits. It contains a lot of data on these contents and their concentrations. However, in my opinion, such presentation within the text makes comparisons analysis impossible. The table containing the information on the quantities of components in different Citrus fruits could be a solution;
Response: Following this interesting question by the Reviewer, we have added a table comparing the hesperidin content of different Citrus fruits (Table 1 in Chapter 2) - I am really interested to know the opinion of the authors what Citrus fruit (and in which form) is the best one;
Response: Following data of Table 1, we have added the following opinion: “Based on these data, it can be said that the choice of the most suitable fruits for a better intake of hesperidin could be made between oranges, mandarins and clementines, according to individual preferences, and costs.” We have added this sentence at the end of the paragraph where the table 1 data is discussed. We limited ourselves to commenting on the quantities of hesperidin and not so much of other components because it is the hesperidin that today considered specifically important for the anti-COVID-19 activity.
- In my opinion short description of the benefits of Citrus fruits in comparison to other fruits should be also included.
Response: We understand the interest of this question but hesperidin is contained only in Citrus fruits and in some other vegetable like peppermint, so we believe that a comparison with other components in other fruits would be beyond the scope of our paper. For the antioxidant capacity, thanks to this question, in the revised paper we have added the only study where a comparison was made between different fruits: “An interesting study compared the antioxidant capacity of the plasma of human subjects after the ingestion of 150 ml of different fruit juices [72]. A significant free radical elimination effect was observed already after 30 minutes and up to 90 minutes after the ingestion of apple, orange, grape, peach, plum, kiwi, melon and watermelon juices, but not of pear juice. The grape juice showed a slightly longer lasting effect (up to 120 minutes after ingestion). No study, however, has evaluated the anti-viral effect of different fruits, but the data reported above in sections 2 and 3 suggest that it is mainly attributable to Citrus fruits due to their distinctive content of hesperidin.”
Reviewer 3 Report
This is an interesting review describing how hesperidin supplementation may be helpful for patients under COVID19. The manuscript is timely, well-written with a clear focus and suits best to the scope of the journal. Please find next some few comments on the paper:
1) The authors should clearly state at the end of the introduction part the novelty of the present work. Although COVID19 pandemic is a new and emerging research area, there are a lot of papers already published. In the context of hesperidin, the authors should clarify their novelty compared to other papers, such as https://www.sciencedirect.com/science/article/pii/S030698772031358X
and
https://www.mdpi.com/2227-9717/8/5/549
2) The introduction part should be more clearly focused on COVID19. In its current short form, it consists of general comments with 5 references not particularly relevant to the topic of interest.
3) What does the subsection 5 add to the review from a mechanistic point of view? Are these effects irrelevant to the redox properties of hesperidin? I think this section has to be more clearly rationalised.
4) The subsection 7 is too long as a "conclusion" section. I would suggest separating "future perspectives" from the general conclusion in different subsections.
Again, I commend the authors for their interesting work.
Author Response
Response to Reviewer 3 Comments
This is an interesting review describing how hesperidin supplementation may be helpful for patients under COVID19. The manuscript is timely, well-written with a clear focus and suits best to the scope of the journal.
Response: We thank the Reviewer for his/her positive comments of to our work
Please find next some few comments on the paper:
- The authors should clearly state at the end of the introduction part the novelty of the present work. Although COVID19 pandemic is a new and emerging research area, there are a lot of papers already published. In the context of hesperidin, the authors should clarify their novelty compared to other papers, such as https://www.sciencedirect.com/science/article/pii/S030698772031358X And https://www.mdpi.com/2227-9717/8/5/549
Response: We thank the reviewer for this question. Actually, we have cited the paper from Haggag et al. in Medical Hypotheses, but it is less informative than ours since we have reviewed many more papers and dedicated an etire chapter to antioxidant components. As for the work of Meneguzzo et al. it reports more recent evidence on the antiviral actions of hesperidin and we have mentioned it extensively, but it is oriented more in a technological sense (extraction and purification of the substance from citrus fruits) than biological and pharmacological, as is ours. Even compared to Meneguzzo et al our text examines the antioxidant and anti-inflammatory effects more extensively and also deals with the vitamin C. The two original figures that enrich our work are much clearer, more complete and explanatory than those presented by other authors.
Following this suggestion of the reviewer, we have enlarged the paragraph of Discussion concerning the hypotheses of anti-COVID-19 action of hesperidin, we have added the citation of Meneguzzo and we have better explained the new contribution of our paper: “The recently accumulated evidence suggests that hesperidin supplementation may be useful as prophylactic agent against SARS-CoV-2 infection and as complementary treatment during COVID-disease, as recently suggested by others [24, 98]. In support of this hypothesis, the latter work cited the in silico study of Wu et al. [31] and proposed that the benefit of hesperidin may derive both from the binding to the coronavirus spike and from its anti-inflammatory activity. As it appears from our review, several other research groups [32-36] have been added to the work of Wu et al [31], showing low binding energy to other viral proteins besides to spike. In this work, we have also given importance to further biological actions of Citrus fruits, in protecting the cell from damage caused by the virus and the oxidative stress. Moreover, from our point of view it is important to consider, in addition to the effect of the single molecule, also the set of benefits of citrus fruits and whole fruit juice. In fact, orange, lemon and mandarin contain a significant amount of hesperidin that can be taken through the diet and contain also vitamin C, which has nutraceutical properties that could synergize with the flavanone.”
- The introduction part should be more clearly focused on COVID19. In its current short form, it consists of general comments with 5 references not particularly relevant to the topic of interest.
Response: We agree with this note and we have made the Introduction much more informative by including part of previous section n. 2, where the infection mechanisms of SARS-CoV-2 are considered, with the possible pharmacological approaches to block them. In Introduction, we have also added a few review references on the different tissue espression of ACE2 receptors: “The internalisation of SARS-CoV-2 is mediated by the binding of the spike glycoprotein of the virus with its receptor (ACE2) on the cell membranes. ACE2 is expressed in several tissues, including alveolar lung cells, gastrointestinal tissue and even the brain [7-10].”
- What does the subsection 5 add to the review from a mechanistic point of view? Are these effects irrelevant to the redox properties of hesperidin? I think this section has to be more clearly rationalised.
Response: We agree with this note and we have made clearer that the benefits of hesperidin in COVID-19 derive not only from the direct antiviral action, but also indirectly from the benefits on general health and prevention of cardiovascular diseases, which in turn may complicate the clinical course of the viral infection. The sequence of paragraphs/concepts in section 5 (6 in revised paper) has been better rationalised.
- The subsection 7 is too long as a "conclusion" section. I would suggest separating "future perspectives" from the general conclusion in different subsections.
Response: We agree with this note and we have separated a “Discussion” (Section 7 in revised paper) from a short conclusion (Section 8 in the revised paper)
Again, I commend the authors for their interesting work.
This manuscript is a resubmission of an earlier submission. The following is a list of the peer review reports and author responses from that submission.